# The Complete Mitochondrial Genome and Gene Arrangement of the Enigmatic Scaphopod *Pictodentalium vernedei*

**DOI:** 10.3390/genes14010210

**Published:** 2023-01-13

**Authors:** Tianzhe Zhang, Yunan Wang, Hao Song

**Affiliations:** 1School of Energy and Environmental Engineering, Hebei University of Technology, Tianjin 300401, China; 2Institute of Oceanology, Chinese Academy of Sciences, Qingdao 266071, China; 3University of Chinese Academy of Sciences, Beijing 101400, China; 4Laboratory for Marine Ecology and Environmental Science, Qingdao National Laboratory for Marine Science and Technology, Qingdao 266237, China; 5Center for Ocean Mega-Science, Chinese Academy of Sciences, Qingdao 266071, China

**Keywords:** mitogenome, *Pictodentalium vernedei*, tusk shells, Scaphopoda, molluscs, Dentaliida, phylogenetic analysis

## Abstract

The enigmatic scaphopods, or tusk shells, are a small and rare group of molluscs whose phylogenomic position among the Conchifera is undetermined, and the taxonomy within this class also needs revision. Such work is hindered by there only being a very few mitochondrial genomes in this group that are currently available. Here, we present the assembly and annotation of the complete mitochondrial genome from Dentaliida *Pictodentalium vernedei*, whose mitochondrial genome is 14,519 bp in size, containing 13 protein-coding genes, 22 tRNA genes and two rRNA genes. The nucleotide composition was skewed toward A-T, with a 71.91% proportion of AT content. Due to the mitogenome-based phylogenetic analysis, we defined *P. vernedei* as a sister to *Graptacme eborea* in Dentaliida. Although a few re-arrangements occurred, the mitochondrial gene order showed deep conservation within Dentaliida. Yet, such a gene order in Dentaliida largely diverges from Gadilida and other molluscan classes, suggesting that scaphopods have the highest degree of mitogenome arrangement compared to other molluscs.

## 1. Introduction

The arrangements of mitochondrial genes appear to have changed very little for some phyla of animals. For instance, the most studied arthropods and vertebrates share near-identical gene arrangements within the phyla. Such q consistency of mitogenomic features in the model animals (fruit flies, mouses and frogs, etc.) has led to a ‘textbook description’ of the conservation mitochondrial genome. Yet, more thorough and extensive mitochondrial genome sampling from animals has discovered numerous instances in which such features are absent. The second largest phylum, Mollusca, is particularly rich in exceptions, and the research in recent years has produced a huge list of unexpected departures from the textbook description in Mollusca, including extreme size variations, radical genome rearrangements, frequent gene duplications and losses, the insertion of novel genes and a bizarre inheritance pattern known as “doubly uniparental inheritance” (DUI) [1]. Thus, it is required for us to continue studying the striking variation in the molluscan mitochondrial genome architecture, molecular function and intergenerational transmission, as these can hold great promise for uncovering animal mitochondrial genome-specific biological processes, histories and functions. Molluscan mitochondrial genomes offer a model system for investigating evolution, the variety of functions that mitochondria serve in organismal physiology and the numerous applications of phylogenetic and population biology that can be derived.

The Scaphopoda is a rare and small clade, consisting of nearly 600 described, extant species [2]. They have a common name, tusk-shell, because of their conspicuous ivory-shaped shell, which opens at both the anterior and posterior ends. The anterior opening is wider, with a burrowing foot that extends out. Unlike most other molluscan relatives, they have no gills or ctenidium, instead they use the mantle for respiration. Scaphopods live infaunal in sand or mud to collect foraminifers, which are their main prey, with numerous captacula and grind them up using their large radula apparatus. Scaphopods are ecologically important bioturbators, and in some circumstances, they can be extremely prevalent in benthic marine communities. For example, in a recent biodiversity assessment of the Prince Gustav Channel in the Weddell Sea, Antarctica [3], the most abundant molluscan class was Scaphopoda, with over 11,000 specimens having been collected in just six epibenthic sledge casts (54% of the molluscan diversity sampled). Scaphopods are also evolutionarily important as they are the closest living relatives of Gastropoda and Bivalvia [4,5,6], the two most diverse and economically important classes of mollusks. Yet, the exact placement of scaphopods within Conchifera has been undecided in terms of molluscan phylogeny. Scaphopods have received much less research attention than most other classes of mollusks have. Most of our knowledge about scaphopod diversity and evolution is from species descriptions and morphological studies [7,8], whereas relatively few pieces of work have studied this group from a molecular perspective. To date, only four sets of mitochondrial genome data have been published in this clade, and it is speculated that the scaphopod mitochondrial genomes have the highest degree of gene re-arrangement compared to those of the other remaining classes of mollusc. To verify this and facilitate a more comprehensive understanding of mitochondrial genome evolution in molluscs, here, we present the complete mitogenome of one representative species, the dentaliid scaphopod, *P. vernedei*.

## 2. Materials and Methods

### 2.1. Sampling and DNA Isolation

An adult animal of *P. vernedei* was sampled in March 2020 from the east China Sea (27°31′32.73″ N, 22°30′39.13″ E). The species identification was carried out by the Department of marine organism taxonomy and phylogeny, IOCAS (Institute of Oceanology, Chinese Academy of Sciences), based on morphology and 16S sequencing. The foot was dissected and preserved in liquid nitrogen. The total genomic DNA was further isolated using CTAB methods.

### 2.2. Mitochondrial Genome Sequencing and Assembling

The whole mitogenome was sequenced by novogene co. ltd, Tianjing, China. In brief, by using the NexteraXT DNA Library Preparation Kit, we constructed the genomic libraries, with an average length of 350 bp, and performed the sequencing of 150 bp paired-end runs on the Illumina Novaseq 6000 platform (San Diego, CA, USA). To meet the quality control, the raw data were further processed by NGS QC Toolkit v2.3.3 as per manual instructions [9]. Then, the de novo assembling of mitochondrial genome was carried out in SPAdes 3.11.0 [10].

### 2.3. Annotation and Characterization of Scaphopod Mitochondrial Genome

The annotation of the *P. vernedei* mitochondrial genome was performed by using MITOS Web Server (http://mitos.bioinf.uni-leipzig.de/index.py, with accession on 12 September 2022). We then used SnapGene Viewer for the manual inspection and corrections of start/stop codons in each gene. The location and length of all of the protein-coding genes (PCGs) were checked under Open Reading Frame Finder (ORFfinder Home, NCBI). The nucleotide composition, composition skew and codon usage of PCGs, as well as the relative synonymous codon usage (RSCU), were analyzed and calculated in PhyloSuite v1.2.0 [11]. We applied the online Proksee tool (https://proksee.ca/, accessed on 27 September 2022) to visualize the mitochondrial genome map. To predict the cloverleaf structures of 22 transfer RNAs, we used tRNAscan-Se [12,13] and MITOS according to the user manual, and we applied ViennaRNA Web Services (http://rna.tbi.univie.ac.at/, accessed on 2 October 2022) to visualize the secondary structures.

To further explore the gene arrangement among the Scaphopoda and other molluscan lineages, we selected 21 complete mitochondrial genomes from multiple lineages (Table 1) and downloaded them from NCBI database with their accession numbers. Of these, 4 species belonged to the Scaphopoda, 3 species belonged to the Protobranchia (presumably the most basal bivalve lineage) and 3 species belonged to vetigastropoda (presumably the most basal gastropod lineages). The *Lingulata* from Brachiopoda was selected as the outgroups.

### 2.4. Phylogenetic Placement

To explore the phylogenetic placement of *P. vernedei* within Scaphopoda, a phylogenetic analysis was carried out as follows. Briefly, 13 protein-coding genes were obtained under PhyloSuite v1.2.2 [11], then, the alignment of their amino acid sequences was performed by Mafft v7 [14]. We used Gblocks 0.91b [15] with default settings to remove those misaligned positions. All of the protein-coding genes were then concatenated into a matrix under PhyloSuite v1.2.2 [11]. We applied ModelFinder [16] with AICc (so-called corrected Akaike Information Criterion, which has a more precise approximation in small samples) to infer a best-fitting partition model. Finally, IQ-Tree v2.1.2 [17] with 1000 ultrafast bootstrap replications was applied to construct a maximum likelihood (ML) tree, which was processed in FigTree v1.4.4 (https://github.com/rambaut/figtree/releases, accessed on 25 December 2022) for visualization.

## 3. Results and Discussion

### 3.1. General Features of Mitochondrial Genome

The total length of the circular mitogenome of *P. vernedei* is 14,519 bp (Figure 1), and this is comparable to those of other four known scaphopod mitogenomes (14,869 of *Antalis entails*, 14,492 of *G. eborea*, 13,932 of *S. lobatum* and 13,790 of *Gadilida* sp.). Molluscs vary largely in their mitochondrial genomes size. The scaphopods, together with heterobranch gastropods (~13.6–14.1 kb) [18,19,20], are among the smallest mitochondrial genomes that have been reported so far [1], while the largest ones come from Arcidae clams (e.g., the *Scapharca kagoshimensis* mitogenome is estimated to be ~56.2 kb) [21]. The *P. vernedei* mitogenome contains 37 mtDNAs that are typical to most metazoans (Table 2 and Figure 1), including thirteen protein coding genes (PCGs), twenty-two tRNAs and two ribosomal RNAs (*12S RNA* and *16S RNA*). Five PCGs (*cox1*, *nad2*, *cytb*, *nad3* and *atp6*) and twelve tRNAs (*tRNA-Ser^TCT^*, *tRNA-Asn*, *tRNA-His*, *tRNA-Gly*, *tRNA-Arg*, *tRNA-Ser^TGA^*, *tRNA-Ala, tRNA-Leu^TAG^, tRNA-Leu^TAA^, tRNA-Glu, tRNA-Trp* and *tRNA-Cys*) are encoded on the heavy strand, whereas the other eight PCGs, ten two tRNAs and two rRNAs are distributed on the light strand. Among these coding genes, a total of 21 intergenic regions were detected, spanning from 1 to 119 bp, wherein the longest one was situated between *12S RNA* and *tRNA-Met*. In contrast, nine overlaps were detected, ranging from 1 to 18 bp. The longest overlapped region was located between *tRNA-Met* and *16S RNA* (18 bp), which was followed by the *cox1* and *tRNA-Ser^TCT^* overlap (14 bp), as well as the *tRNA-His* and *cox2* overlap (14 bp).

The nucleotide composition was 36.46% for A, 35.45% for T, 13.24% for C and 14.85% for G in *P. vernedei*, and as a whole, it was biased toward A+T content, at 71.91%. In other Dentaliida species, the A+T contents exceed 70%, indicating that the Dentaliida mitogenomes are more biased toward A+T (Table 3). In addition, *P. vernedei* has a positive total A-T skew and a negative G-C skew. Among the eleven molluscs plus two outgroups that were analyzed in present study, positive A-T skews were found in six species, while negative G-C skews were found in nine species (Table 3), showing no apparent consistency.

### 3.2. Protein-Coding Genes (PCGs)

*P. vernedei* has 13 PCGs in its mitochondrial genome, whose length is 10,842 bp in total, and this accounts for 74.67% of the complete mitogenome. Their size ranges from 159 (*atp8*) to 1647 bp (*nad5*), and this is generally similar to most other molluscs, for example, *E. rumphii* [22]. In *P. vernedei*, these PCGs use four different triplets as the start codon, including ATG, ATT, TTG and ATA. Such a usage is similar to its sister Dentaliida *G. eborea*, which also uses ATG, ATT, TTG and ATA as a start codon, but not TTG [23]. In contrast, in other molluscs such as some vetigastropods, ATG is the only start codon used for all of the PCGs [22]. These 13 PCGs contain various types of stop codons; TAA was found in seven PCGs (*cox1*, *cox2*, *cox3*, *nad3*, *nad4*, *nad4l* and *nad5*), while TAG was found in three PCGs (*atp8*, *cytb* and *nad6*) and T was found in the remaining three PCGs (*atp6*, *nad1* and *nad2*). Using TAA and TAG as stop codons is widely common in molluscs, and using T is also prevalent in scaphopods such as *G. eborea* [23]. 

In terms of the base composition of PCGs, T accounts for most (42.88%), followed by 28.24% of A, 15.22% of C, and 13.66% of G. The A-T content in both the individual PCG and concatenated matrices surpass 60%, suggesting that A-T is more preferred in *P. vernedei* PCGs. The A-T skews in all of the PCGs are negative, indicating they are biased toward T. With regard to the G-C skew, eleven PCGs show a positive G-C skew, whereas the remaining two PCGs (atp6 and nad6) show a negative one.

As shown in Figure 2, the distribution of relative synonymous codon usage (RSCU) of *P. vernedei* mitogenome indicates that Leu, Phe and Ile are the top three most frequently utilized amino acids, while UUA-Leu, UCU-Ser2, ACU-Thr and CCU-Pro are the top four most frequently used codons. Among these twenty-two amino acids, nine of them (i.e., Ala, Arg, Gly, Leu1, Pro, Ser1, Ser2, Thr and Val) are encoded by four codons, while the others are encoded by two codons, which is common in other molluscs.

Annotating genes in the molluscan mitochondrial genome is notoriously challenging. While it is easy to identify the protein-encoding genes by searching the similarity of orthologous counterparts, it could be difficult to infer the valid start codon and stop codon. This is partly because molluscs usually have multiple alternatives for both start and end codons [1]. For instance, in addition to ATG, start codons in molluscs also include TTG, ATY, ATA, and GTG (which are responsible for leucine, isoleucine, methionine, and valine encoding, respectively). Each of them has a match to the trnM anticodon for at least two nucleotides, suggesting their dual role in mitochondria, that is, they serve as the transfer RNA for both methionine and, as for protein initiation, formyl-methionine. For the stop codon, in some cases, molluscs do not just terminate at a complete TAA or TAG stop codon but terminate on just a TA or T that is completed to a TAA stop by polyadenylation of the transcript. Another thing that aggravates the difficulty in inferring start and stop codons is the pervasive gene overlapping with the upstream/downstream gene, even when on the same strand. This requires us to take a full consideration of the extent of the ORF evolutionary conservation and manually check the start/stop codon carefully.

### 3.3. Ribosomal RNAs and Transfer RNAs

There are two ribosomal RNAs: one is 12S ribosomal RNA (*rrnS*), with a length of 636 bp, and another one is 16S ribosomal RNA (*rrnL*), with a length of 1200 bp. The former one, *rrnS*, is located between *tRNA-Met* and *tRNA-Thr*, while the latter one, *rrnL*, is located between *tRNA-Val* and *tRNA-Met*. For *rrnS*, the A-T skew and G-C skew are 0.034 and 0.098, respectively; for *rrnL*, they are 0.057 and 0.22. Both two ribosomal RNAs are biased toward T and G.

*P. vernedei* possesses 22 transfer RNA genes, with their size ranging from 59 (*tRNA-Arg*) to 71 bp (*tRNA-Thr*). The A-T content in the total transfer RNAs is 75.5%, which also exhibits an obvious A-T bias. Transfer RNAs usually have typical clover-leaf secondary structures, yet in *P. vernedei*, truncated Di-hydroxy uridine (DHU) arms without paired bases occur in two tRNAs, *tRNA-Arg* and *tRNA-Ser1* (Figure 3). The DHU arm of *tRNA-Ser1* is still eleven bases long, but that of the tRNAs Arg has been reduced to five bases. In its close sister species *G. eborea*, five tRNAs have DHU arms, including tRNA-Ser1, tRNA-Arg, tRNA-Thr and tRNA-Ile [23]. Thus, DHU lost in tRNA-Ser1 and tRNA-Arg can be seen as a synapomorphy in this clade. In addition, for tRNA-Ser1, the DHU loss can be widely founded in other metazoan mitogenomes [24], suggesting it is either an ancestral feature or a convergent consequence.

### 3.4. Phylogenetic Placement

To understand the placement of *P. vernedei* and the relationship between the scaphopods, bivalves and gastropods, we analyzed three bivalves, three gastropods, five current available scaphopods and two outgroups (*Platynereis* and *Lingula*). The phylogenetic analysis was performed based on the alignment of 13 protein-coding genes (Figure 4A). In the present study, the scaphopods were grouped into a monophyletic clade, within which *P. vernedei* was placed as a sister to *G. eborea* with BS = 95. Together with *A. entalis*, they form a clade with full support, representing Dentaliida. We also recovered the monophyletic Gadilida containing *S. lobatum* and *Gadilida* sp. with full support. However, the bootstrap support value between Dentaliida and Gadilida is not high. This is partly because the current matrix contains insufficient phylogenetic signals to resolve this node. We argue that more taxa sampling is needed for revising the scaphopod internal systematics.

The phylogenetic placement of Scaphopoda within Conchifera remains to be a notorious problem. There are at least six competing hypotheses showing discrepancies in scaphopod placement: the Scaphopoda can be placed as sister to (1) Bivalvia in ‘Diasoma’ concept [25,26,27], (2) Cephalopoda in the ‘helcionellid’ concept [28,29,30], (3) Gastropoda [6,31,32], (4) Gastropoda-Cephalopoda in the Haszprunar hypothesis or (5) Pleistomollusca (Bivalvia+ Gastropoda) [4]. Though Cephalopoda and Monoplacophora mitogenomes were not included in present study, we found that Bivalvia and Gastropoda form a clade (Pleistomollusca), and Scaphopoda is a sister to it, with BS = 75. The transcriptome-based phylogenomics results are consistent with the previous findings from large-scale transcriptome-based phylogenomics [4]. However, we hoped that the complete mitochondrial genomes would provide new character sets to help resolve the molluscan relationships, but it is now apparent that molluscan mitogenomes did not provide the useful resolution as they were expected to [2,33]. Even when we were using whole mitogenomes data sets, gastropods can be diphyletic. Mitogenomes can fail to resolve deep relationships of molluscan major lineages, partly because mitogenome has a more accelerated evolutionary rate than the genomes for most metazoans so [34], and more importantly, because molluscan mitogenomes have a surprisingly wide variation in size, rearrangements, duplications/losses and in the introduction of novel genes [1]. Such diverse changes are severally biased to different lineages, making it not a good tool for resolving deep molluscan relationships. We thus argue to use whole genome data to confidently resolve the placement of Scaphopoda in the future.

### 3.5. Mitochondrial Gene Arrangement

Molluscs break the rules of mitochondrial genomes, as they are one of the few metazoan groups that show extensive gene rearrangement. Such prevalent rearrangements in this phylum can hamper the reconstruction of relationships as discussed above, but on the other hand, such variation occurring among the molluscan lineages is also helpful in understanding the underlying mechanisms of gene reordering. Here, we compared four scaphopod gene arrangements (*Gadillida* sp. was not included because of its incomplete assembly and annotation, and only these four are currently available and complete) with three protobranch bivalves and three vetigastropods (Figure 4B) because Protobranchia and Vetigastropoda showed deep conservation in the mitochondrial gene order, with only a few steps of rearrangements occurring comparing to those for chitons and other lophotrochozoans, thus, these are supposed to be most approximal to the ancestral states [2].

We found that *P. vernedei* shared an identical gene order with dentaliid sister *G. eborea*, yet in another dentaliid *A. entails* there is an inversion region, namely, *trnF-trnK-nad5-trnD-nad4-nad4L*. Compared to Protobranchia and Vetigastropoda, the gene order in *P. vernedei* is highly re-arranged, with only one region (*nad6-trnP-nad1*) showing the same pattern. Such neighboring relationships between those three blocks can also be seen in Cephalopoda and Monoplacophora [35], indicating that this order is a symplesiomorphy in Conchifera. Surprisingly, in gadilid *S. lobatum*, the gene order is completely divergent, and it has lost the conserved *nad6-trnP-nad1* relationship and shows no conservative region with both the dentaliid groups and other molluscan lineages. This suggests extensive gene rearrangement not only occurred when Scaphopoda split from other lineages, but it also occurred in the split between Gadilida and Dentaliid.

## 4. Conclusions

Our study presents a complete mitogenome of *P. vernedei,* whose size was 14,519 bp. It contains typical thirty-seven mtDNAs, including thirteen PCGs, twenty-two tRNAs and two rRNAs. The arrangement of the mitochondrial genes is identical to those of dentaliid *G. eborea*, but it is quite divergent from those of Gadilida and other molluscan lineages, suggesting that Scaphopoda has gone through substantial gene rearrangement. The extensive rearrangement in scaphopods implies that there has been little conservation, thus making it not a good choice for resolving the placement of Scaphopoda among the molluscs. Given that previous phylogeny analyses using morphological data, transcriptome data, and mitogenome data all leave the scaphopod placement undetermined, we argue for the use of whole genome data, which comprise the most comprehensive genetic information to resolve this problem in the future. Since the scaphopod showed more frequent gene rearrangement than other molluscan lineages did, it provides a model system to study how such substantial rearrangements are in concert with animal physiology and adaptative evolution.

## Figures and Tables

**Figure 1 genes-14-00210-f001:**
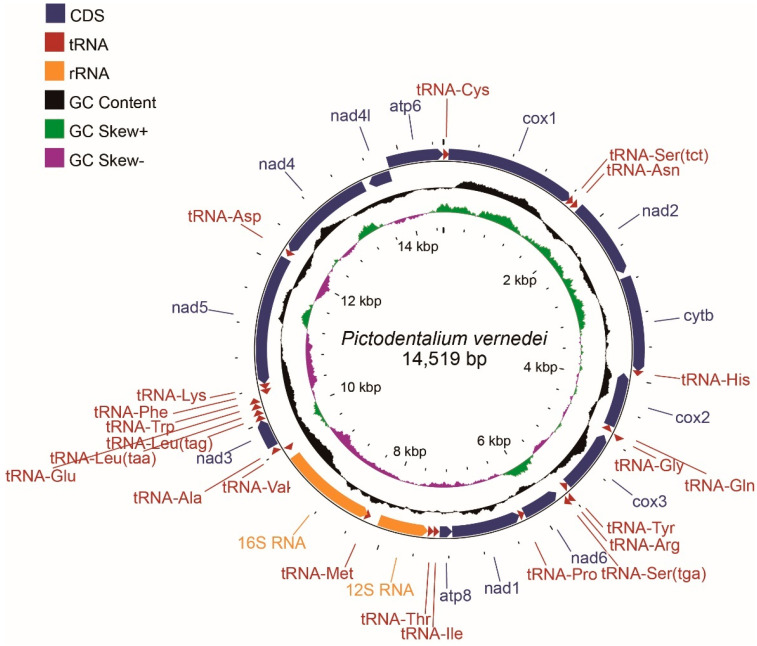
The overall mitogenome map of *P. vernedei*. From the inside to the outside, the innermost scale indicates the length, the inner circle indicates the G-C skew, the middle circle indicates the G-C content, and the outer circle indicates gene arrangement. The G-C skew was centered at zero while the G-C content circus was centered at 50%.

**Figure 2 genes-14-00210-f002:**
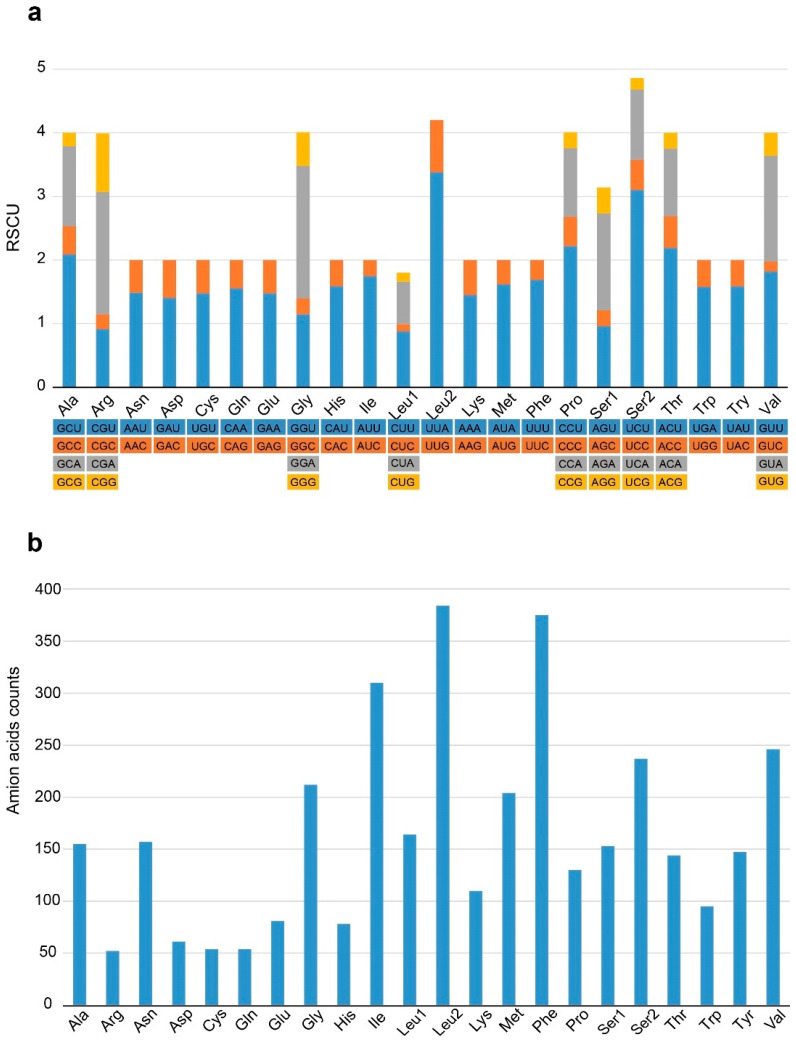
Distribution of (**a**) relative synonymous codon usage (RSCU) and (**b**) amino acids frequency in *P. vernedei* mitogenome.

**Figure 3 genes-14-00210-f003:**
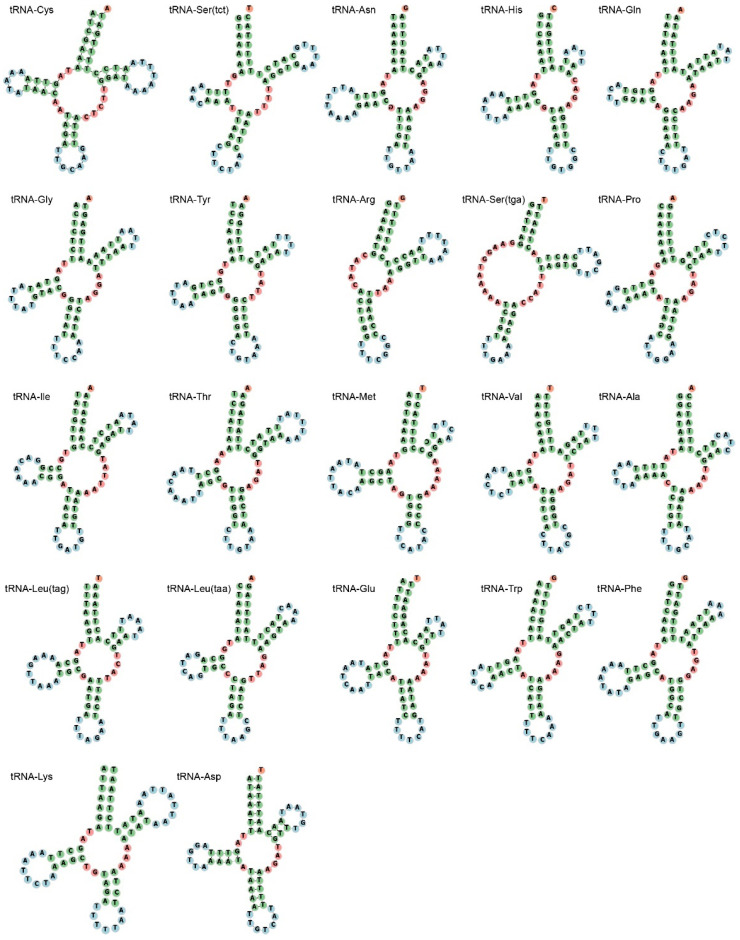
Cloverleaf structures of the 21 tRNA genes of *P. vernedei* mitochondrial genome.

**Figure 4 genes-14-00210-f004:**
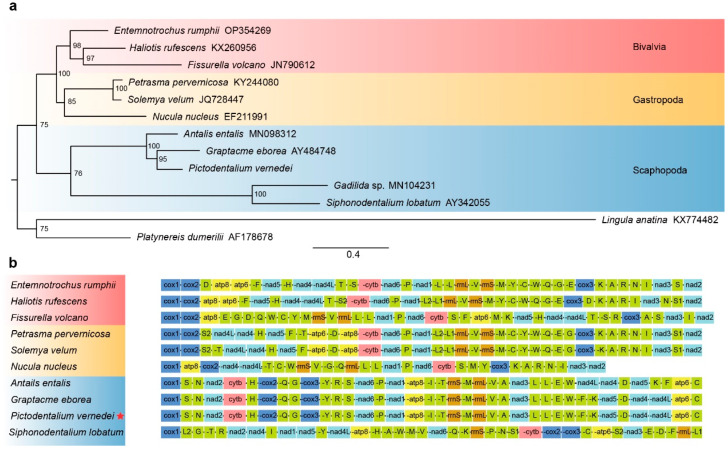
Phylogenetic tree (**a**) and gene orders (**b**). (**a**) Three protobranch bivalves, three vetigastropods, five currently available scaphopods, and 2 outgroup species were selected, and the phylogenetic tree was constructed based on the amino acid sequences from 13 protein-coding genes. Bootstrap (BS) support values are shown at each node. (**b**) Gene order of PCGs among Vetigastropoda, Protobranchia, and Scaphopoda. *Gadillida* sp. was not included as its assembly and annotation are not complete. The arrangement graph is visualized by iTOL and the boxes with different colors and labels represent different gene blocks.

**Table 1 genes-14-00210-t001:** Species selected for phylogenetic analysis and gene order comparison.

Phylum	Class	Subclass	Order	Species	Accession Number
Mollusca	Gastropoda	Vetigastropoda	Pleurotomariida	*Entemnotrochus rumphii*	OP354269
			Lepetellida	*Haliotis rufescens*	KX260956
				*Fissurella volcano*	JN790612
	Bivalvia	Protobranchia	Solemyida	*Petrasma pervernicosa*	KY244080
				*Solemya velum*	JQ728447
			Nuculida	*Nucula nucleus*	EF211991
	Scaphopoda		Dentaliida	*Antalis entalis*	MN098312
				*G. eborea*	AY484748
				*P. vernedei*	This study
			Gadilida	*Siphonodentalium lobatum*	AY342055
				*Gadilida* sp.	MN104231
Annelida	Polychaeta	Errantia	Phyllodocida	*Platynereis dumerilii*	AF178678
Brachiopoda	Lingulata		Lingulida	*Lingula anatina*	KX774482

**Table 2 genes-14-00210-t002:** Organization of the coding genes in *P. vernedei* mitochondrial genome.

Gene Name	Location		Length (bp)	Codon		Anticodon	Intergenic Region	Strand
	Start	End		Start	Stop			
tRNA-Cys	1	68	68			GCA	0	H
cox1	63	1619	1557	TTG	TAA		−6	H
tRNA-Ser(tct)	1606	1670	65			TCT	−14	H
tRNA-Asn	1675	1740	66			GTT	4	H
nad2	1761	2700	940	ATG	T		20	H
cytb	2755	3888	1134	ATA	TAG		54	H
tRNA-His	3879	3940	62			GTG	−10	H
cox2	3927	4616	690	ATG	TAA		−14	L
tRNA-Gln	4617	4680	64			TTG	0	L
tRNA-Gly	4687	4753	67			TCC	6	H
cox3	4756	5535	780	ATG	TAA		2	L
tRNA-Tyr	5536	5600	65			GTA	0	L
tRNA-Arg	5599	5657	59			TCG	−2	H
tRNA-Ser(tga)	5657	5720	64			TGA	−1	H
nad6	5714	6199	486	ATG	TAG		−7	L
tRNA-Pro	6200	6264	65			TGG	0	L
nad1	6265	7147	883	ATG	T		0	L
atp8	7148	7306	159	ATG	TAG		0	L
tRNA-Ile	7314	7381	68			GAT	7	L
tRNA-Thr	7384	7454	71			TGT	2	L
12S RNA	7456	8091	636				1	L
tRNA-Met	8211	8276	66			CAT	119	L
16S RNA	8259	9458	1200				−18	L
tRNA-Val	9549	9611	63			TAC	90	L
tRNA-Ala	9621	9683	63			TGC	9	H
nad3	9699	10,037	339	TTG	TAA		15	H
tRNA-Leu(tag)	10,039	10,103	65			TAG	1	H
tRNA-Leu(taa)	10,105	10,169	65			TAA	1	H
tRNA-Glu	10,179	10,243	65			TTC	9	H
tRNA-Trp	10,248	10,313	66			TCA	4	H
tRNA-Phe	10,322	10,389	68			GAA	8	L
tRNA-Lys	10,390	10,458	69			TTT	0	L
nad5	10,458	12,104	1647	ATT	TAA		−1	L
tRNA-Asp	12,143	12,207	65			GTC	38	L
nad4	12,209	13,477	1269	ATT	TAA		1	L
nad4l	13,549	13,839	291	ATT	TAA		71	L
atp6	13,853	14,519	667	ATT	T		13	H

**Table 3 genes-14-00210-t003:** Statistics of nucleotide composition in selected molluscan mitogenomes.

Species Selected	Size (bp)	A%	T%	G%	C%	A-T Skew	G-C Skew	AT Proportion
*E. rumphii*	15,795	35.21	29.98	14.43	20.39	0.08023	−0.17112	65.19%
*H. rufescens*	16,646	35.39	24.93	13.75	25.93	0.17339	−0.30719	60.32%
*F. volcano*	17,575	25.72	35.83	26.69	11.76	−0.16431	0.38819	61.55%
*P. pervernicosa*	16,554	34.45	31.84	12.64	21.07	0.03946	−0.24996	66.29%
*S. velum*	15,660	35.53	32.58	13.14	18.74	0.04341	−0.17565	68.11%
*N. nucleus*	13,671	32.13	31.78	15.66	20.42	0.00549	−0.13197	63.92%
*A. entalis*	14,869	35.30	40.41	12.39	11.90	−0.06742	0.01993	75.71%
*G. eborea*	14,492	36.98	37.14	12.67	13.21	−0.00214	−0.02081	74.12%
*P. vernedei*	14,519	36.46	35.45	13.24	14.85	0.01408	−0.05738	71.91%
*S. lobatum*	13,932	32.08	36.20	18.92	12.81	−0.06036	0.19258	68.27%
*Gadilida* sp.	13,790	32.08	32.82	14.35	20.74	−0.01134	−0.18214	64.91%
*P. dumerilii*	15,619	31.22	32.92	15.41	20.45	−0.02646	−0.14071	64.14%
*L. anatina*	24,876	26.28	36.53	21.29	15.90	−0.16308	0.14483	62.81%

## Data Availability

The mitochondrial genome of *P. vernedei* is available from GeneBank under the accession OQ198438.

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
