# Peer review of "The Complete Mitochondrial Genome and Gene Arrangement of the Enigmatic Scaphopod Pictodentalium vernedei"

_genes, 2023, doi:10.3390/genes14010210_

Round 1

Reviewer 1 Report

Happy New Year, at the beginning of this year I am happy to be a reviewer of the potential manuscript "The Complete Mitochondrial Genome and Gene Organization of the Enigmatic Scaphopod Pictodentalium vernedei (Mollusca)"

As the authors state, to date only 4 mitochondrial genome data have been published in this clade, and it is speculated that scaphopod mitochondrial genomes have the highest degree of gene rearrangement, compared to other remaining classes of molluscs. To verify this and facilitate a better understanding of mitochondrial genome evolution in molluscs, the manuscript presents the complete mitochondrial genome of one representative species, the dentaliid scaphopod Pictodentalium vernedei.

Following are my concerns that need to be addressed by authors:

1. The title should be without "(Mollusca)".

2. To improve readability, add the following keywords: tusk shells, molluscs, Dentaliida.

3. On the method, 2.1. Sampling and DNA Extraction: Please provide details on how authentication and identification are carried out? by whom and in which laboratory?

4. Finally, lots of abbreviations. Therefore, it is necessary to provide a list of abbreviations.

Author Response

Dear reviewer:

We are very grateful to your comments for the manuscript. According with your advice, we tried our best to amend the relevant part and made some changes in the manuscript. These changes will not influence the content and framework of the paper. All of your questions were answered below. And here we list the changes and marked in red in revised paper.

1. The title should be without "(Mollusca)".

•Now removed.

2. To improve readability, add the following keywords: tusk shells, molluscs, Dentaliida.

•These keywords are now added.

3. On the method, 2.1. Sampling and DNA Extraction: Please provide details on how authentication and identification are carried out? by whom and in which laboratory?

•We have included this information in method part.

4. Finally, lots of abbreviations. Therefore, it is necessary to provide a list of abbreviations.

•We have included a list of abbreviations at the end of the main text.

Reviewer 2 Report

Only few studies exist on the mitochondrial genomes of molluscs, and especially of the scaphopods.

Results from the present paper may help to clarify the taxonomy of this class of molluscs, which still needs revision.

The experiments are carefully done, and the manuscripts is professionally written.

It needs only few misspelling corrections.

Author Response

Thank you for your positive feedback, which is much appreciated. We have inspected and corrected the misspelling.